# Structure of a MacAB-like efflux pump from *Streptococcus pneumoniae*

Hong-Bo Yang [1,2], Wen-Tao Hou[1,2], Meng-Ting Cheng[1,2], Yong-Liang Jiang[1,2], Yuxing Chen[1,2] & Cong-Zhao Zhou [1,2]

The *spr0693-spr0694-spr0695* operon of *Streptococcus pneumoniae* encodes a putative ATP-binding cassette (ABC)-type efflux pump involved in the resistance of antibiotics and anti-microbial peptides. Here we report the crystal structures of Spr0694–0695 at 3.3 Å and Spr0693 at 3.0 Å resolution, revealing a MacAB-like efflux pump. The dimeric Spr0694–0695 adopts a non-canonical fold of ABC transporter, the transmembrane domain of which consists of eight tightly packed transmembrane helices with an insertion of extracellular domain between the first and second helices, whereas Spr0693 forms a nanotube channel docked onto the ABC transporter. Structural analyses combined with ATPase activity and antimicrobial susceptibility assays, enable us to propose a putative substrate-entrance tunnel with a lateral access controlled by a guard helix. Altogether, our findings provide structural insights and putative transport mechanism of a MacAB-like efflux pump in Gram-positive bacteria.

[1] Hefei National Laboratory for Physical Sciences at the Microscale and School of Life Sciences, University of Science and Technology of China, Hefei, Anhui 230027, China. [2] Key Laboratory of Structural Biology, Chinese Academy of Science, Hefei, Anhui 230027, People's Republic of China. Hong-Bo Yang, Wen-Tao Hou and Meng-Ting Cheng contributed equally to this work. Correspondence and requests for materials should be addressed to Y.-L.J. (email: jyl@ustc.edu.cn) or to Y.C. (email: cyxing@ustc.edu.cn) or to C.-Z.Z. (email: zcz@ustc.edu.cn)

ATP-binding cassette (ABC) transporters, which have been found in all kingdoms of life, are involved in diverse physiological activities through the transport of various molecules[1]. According to the transport direction, ABC transporters are divided into importers and exporters. Importers, which exist in prokaryotes and plants[2, 3], facilitate the uptake of small molecules or macro biomolecules such as ions, sugars, amino acids, short peptides, oligosaccharides and metal chelates[4,5]. Exporters have been found in both prokaryotes and eukaryotes, and are capable of extruding diverse substrates including poly-(ribitol-phosphate) polymer of cell wall assembly[6], phospholipids and cholesterol[7]. Particularly, exporters represent one of the main families of efflux pump that are involved in transferring endotoxin[8] or xenobiotics[9]. For instance, the multi-drug efflux pump Sav1866 from *Staphylococcus aureus* is reported to transport ethidium bromide[10], whereas human multidrug efflux pump ABCB1 contributes to multidrug resistance to adriamycin in carcinoma cells[11].

ABC transporters usually contain two transmembrane domains (TMDs) and two cytosolic nucleotide-binding domains (NBDs), in addition to an extra substrate-binding protein for importers. Hydrolysis of ATP by the NBD triggers the conformational changes in TMD, resulting in the transport of substrate across the membrane. The canonical importers commonly contain 10–20 transmembrane (TM) helices, whereas exporters usually have 12 TM helices. Most ABC transporters adopt an alternating access mechanism, namely transporting substrates across the membrane via repetitive switch of inward-facing and outward-facing conformations[12]. However, the structure of lipid-linked oligosaccharide flippase PglK proposed a different mechanism, termed outward-only, in which, only the TMDs at the outward-facing, but not the inward-facing, conformation bind the substrate[13]. In addition, the recent structure of human lipid exporter ABCA1 supplemented a lateral access mechanism, in which substrate enters from the membrane inner leaflet[7].

*Streptococcus pneumoniae* is a Gram-positive pathogen, which is the main cause of acute pneumonia, otitis media, and meningitis in humans[8]. The operon *spr0693-spr0694-spr0695* of *S. pneumoniae* R6 encodes a putative ABC transporter that contributes to the resistance of antibiotics and antimicrobial peptides[14]. Sequence analyses suggest that the Spr0693–0694–0695 assembly is similar to *Escherichia coli* MacAB, with Spr0693 and Spr0694–0695 corresponding to MacA and MacB, respectively. Spr0693 shares a sequence identity of 21% with MacA, whereas Spr0694–0695 complex is 37% sequence identical to MacB. These ABC transporters belong to the subfamily 3.A.1.122, which are termed as macrolide exporters in the prokaryotic ABC3 family[15]. The prominent examples of this subfamily are YknYZ and MacAB, both of which are non-canonical ABC transporters consisting of only eight TMs. Moreover, they usually cooperate with other partner proteins to fulfill the efflux function. For instance, YknYZ functions with YknX to achieve the efflux of substrates[16], whereas MacAB assembles with TolC to form an ABC-type multidrug efflux pump that mediates the transport of macrolides[17] or enterotoxin out of Gram-negative bacteria[18,19]. The recent cryo-electron microscopy (cryo-EM) structure of *E. coli* MacAB-TolC assembly revealed that MacB is a non-canonical ABC transporter with tightly packed 4 + 4 TMs[19]. The periplasmic domain of MacB interacts with a membrane fusion protein (MFP) MacA, which further assembles with TolC to form an elongated tunnel for exporting substrate out of the outer membrane. Despite the structures and molecular functions of a series of tripartite efflux pumps in Gram-negative bacteria have been elucidated[20–22], no structures of the evolutionary counterparts in Gram-positive bacteria have been reported to date. Moreover, the transport mechanism for these non-canonical ABC transporters remains unclear due to the very few structural information.

Here we present the crystal structures of *S. pneumoniae* Spr0694–0695 and Spr0693, representing the structure of a MacAB-like efflux pump in Gram-positive bacteria. Structural analyses reveals that Spr0694–0695 is featured with an atypical TMD, which contains a putative guard helix to control the lateral-access substrate entrance. Moreover, similar to MacA, Spr0693 also forms a hexameric nanotube channel, and docks to the transporter Spr0694–0695 to form an expanded pump that crosses the inner cell-wall space. Our findings provide insights into the assembly and lateral substrate entry of this family of ABC transporters. However, more investigations are needed to further explore the physiological functions of this efflux pump.

## Results

**Structure of a non-canonical ABC transporter Spr0694–0695.** The crystal structure of Spr0694–0695 was determined at 3.3 Å resolution. Each asymmetric unit contains an ABC transporter with a pseudo two-fold symmetry, the NBD and TMD of which are encoded by *spr0694* and *spr0695*, respectively. Spr0695/TMD could be further divided into two domains, a 4-TM helical bundle and an extracellular domain (ECD) inserted between TM1 and TM2. The whole complex is almost 130 Å in length spanning from the cytoplasm to extracellular space, with a height of 44 Å for the intracellular NBDs, 32 Å for the transmembrane TMDs and 52 Å for the extracellular ECDs (Fig. 1a).

Notably, each TMD subunit has only four transmembrane (TM) helices, which is distinct from the TMD of canonical exporters that usually possesses six TM helices. The two discrete 4-TM bundles are folded in a back-to-back manner, without domain swapping usually applied in other ABC transporters, especially in type I exporters[1]. TM2 stretches down into the cytoplasm to form the coupling helix (residues Thr319-Leu328) binding to the NBD. It is distinct from most exporters, of which the coupling helices are swapped from the symmetric subunit owing to the crossover of the flanking TM helices. The helices TM1 and TM2, together with their symmetric counterparts, pack against each other to form the transmembrane hydrophobic core, whereas helices TM3 and TM4 surround at the lateral sides. Of note, both TM1 and TM2 protrude out of the membrane, and bend away from the symmetric axis, resulting in a kink at the extracellular face, thus form a "Y-shaped" conformation together with their counterparts in the symmetric subunit. In contrast, the TMDs buried in the membrane pack against each other tightly via hydrophobic interactions with a buried interface of ~1800 Å$^2$, leaving no opening to the intracellular face of the membrane.

The insertion (residues Asn56–Glu265) between TM1 and TM2 forms an extracellular domain, namely ECD. Similar to the MacB periplasmic domain (PLD)[23], the ECD can also be divided into two subdomains: EST and ESM, named after the extracellular subdomain N-terminal, C-terminal domain and extracellular subdomain αβ mixed domain, respectively (Fig. 1b). The EST is composed of both the N- and C-termini of the ECD, folding into a three-stranded β-sheet packing against two α-helices, whereas the ESM, which consists of two β-sheets (β3–β4–β8 and β5–β6–β7–β9–β10) wrapped by four α-helices, donates most contacts with the TM helices. Notably, a short helix α6 (residues Pro204–Tyr212) of the ESM, which is located at the interface between the ECD and TMD, sticks to the TM helices via hydrophobic interactions and four salt bridges: Lys207 and Lys210–Glu374, Arg208-Asp49 and Glu205–Lys52 (Fig. 1c).

Spr0694 adopts a classic NBD fold of ABC transporter, with an α-helical subdomain and a RecA-like ATPase core subdomain of eight-stranded β-sheet[24]. The two NBD subunits are somewhat

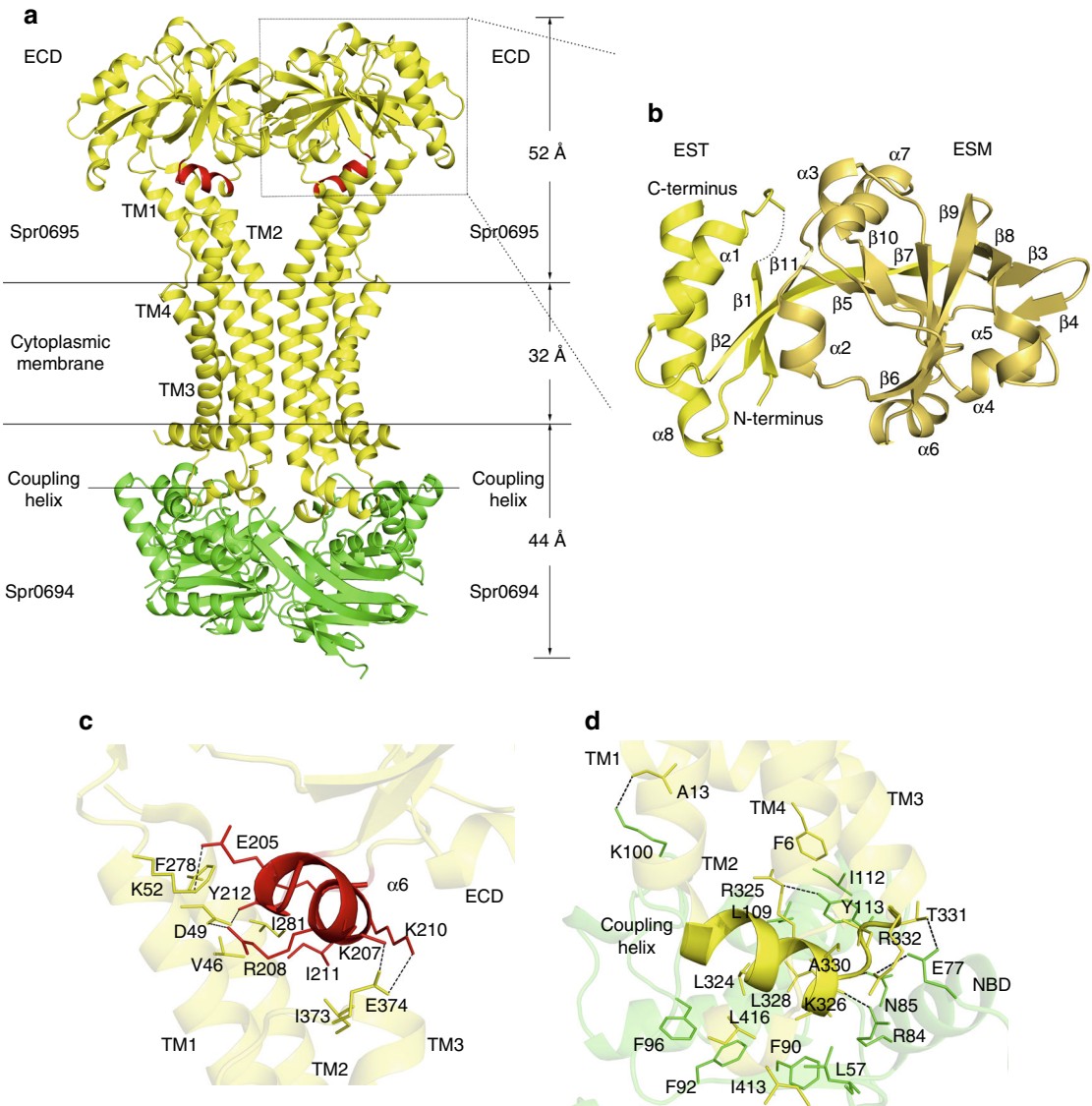

**Fig. 1** Structure of Spr0694–0695. **a** Overall structure of Spr0694–0695.The NBDs and TMDs are colored in green and yellow, respectively. A short helix at the interface of TM helices and ECD is colored in red. **b** Overall structure of the ECD, with the secondary structural elements labeled sequentially. The two subdomains of the ECD (the EST and the ESM) are colored in yellow and yellow-orange, respectively. **c** Interactions between the helix α6 and TMD. The interacting residues are shown as sticks. The helix α6 from the ECD, mediating most contacts with TM helices, is colored in red. **d** Interactions between the coupling helix and the NBD

separated, leaving the ATP-binding pocket exposed to the solvent. However, the nucleotide is absent from the electron density map, despite the protein was co-crystallized with ATP analog AMP-PNP. An interdomain hydrogen bond is formed between Glu203 of one NBD subunit and Thr77′ of the symmetric one, and vice versa. Remarkably, the signature coupling helix of the TMD is embedded in a groove of the NBD in the same symmetric unit, and contributes to the majority of contacts between the TMD and the NBD (Fig. 1d). The groove is in general hydrophobic thanks to the surrounding residues Ala330, Ile112, Leu109, Phe96, Phe92, Phe90, and Leu57. The hydrogen bonds of Arg325–Tyr113, Lys326–Arg84, Leu328–Asn85, and Thr331–Glu77, in addition to a pair of salt bridge Arg332–Glu77 further stabilize the interface between the coupling helix and the NBD.

**The hexameric Spr0693 forms a nanotube channel.** As initial efforts to get the full-length structure of Spr0693 failed after exhaustive trials, we made a series of truncations and finally a

truncated version (Ser59–Lys324) without the membrane-proximal (MP) domain, enabled us to solve the structure at 3.0 Å. Similar to the known MFP structures of AcrA[20] and MacA[21] from Gram-negative bacteria, each subunit of our truncated Spr0693 can be divided into three distinct domains: the α-helical hairpin domain, the lipoyl domain and the β-barrel domain (Fig. 2a). The quaternary structure of Spr0693 displays a typical hexameric nanotube (Fig. 2b), similar to the previously reported MFP structures[20–22]. The two outermost α-helices form a twisted left-handed α-helical hairpin domain; and six helical hairpins further pack into a cylindrical assembly to form an α-helical barrel of 31 Å in diameter and 56 Å in height (Fig. 2b). The 24-Å high lipoyl domain at the middle harbors two layers of three-stranded β-sheets packing against each other, with a topology similar to the biotinyl/lipoyl carrier proteins and domain family[25]. The β-barrel domain at the bottom is composed of six twisted anti-parallel β-strands and a short α-helix, which form a stacked β-barrel of 24 Å in height. The six lipoyl domains,

as well as the six β-barrel domains, are aligned in a head-to-tail manner to form a double-layered ring-like structure in the quaternary structure.

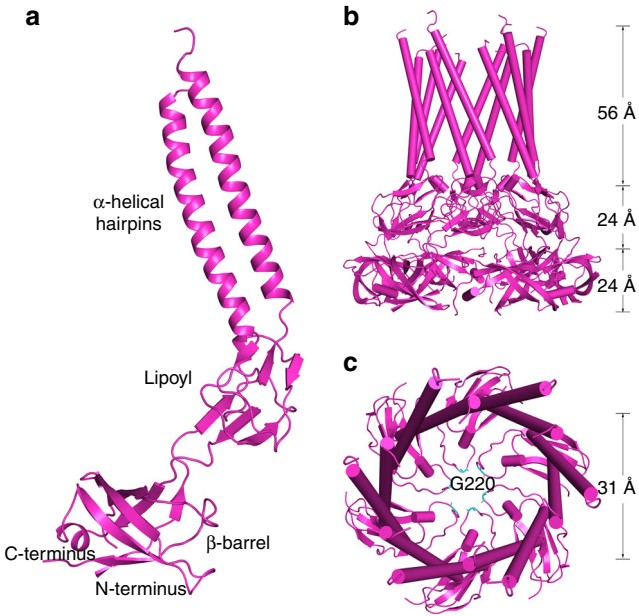

**Fig. 2** Structure of Spr0693. **a** The Spr0693 monomer. The three domains, namely α-helical hairpins, lipoyl and β-barrel, are marked. **b** Side-view of Spr0693 hexamer. Six α-helical hairpins, shown in cylindrical helices, pack against each other to form an α-helical barrel. **c** Top-view of Spr0693 from the α-helical barrel, with a diameter of ~31 Å. A gating ring is formed by the flexible loops from the lipoyl domains, with six Gly220 residues at the innermost site

A top view of the Spr0693 cylinder structure reveals a narrow pore, which is formed by the loops (residues Asn211–Val224) from the lipoyl domains (Fig. 2c). These loops adopt a so-called turn-back conformation and point towards the center, forming a hexameric ring with the residue Gly220 at the innermost site. The pore is likely constrained by the six loops, giving an internal diameter of ~10 Å in our structure. However, the glycine residue Gly220, as well as the loop, has a relatively higher B-factor, and the primary sequence of the loop is variable in the homologs of Spr0693, suggesting that the ring is structurally flexible and might have a rather broad spectrum of substrate selectivity. Nevertheless, the variable ring should be important for substrate transport, as the counterpart loops in MacA were assumed to form a gating ring to prevent backflow and favor the substrate translocation in an outward direction[19].

**Spr0693 augments the ATPase activity of Spr0694–0695.** Previous report suggests that the operon *spr0693–0694–0695* encodes a putative efflux pump[14], indicating that Spr0693 might assemble with Spr0694–0695 to form an intact complex. To confirm that the three proteins Spr0693, Spr0694 and Spr0695 form a complex in vitro, we performed the pull-down assays using the His-tagged Spr0694–0695 and Strep-tagged Spr0693 as the bait. After tandem-affinity purification, the eluted proteins were applied to gel electrophoresis and the corresponding bands were further identified by mass spectrometry (Fig. 3a, Supplementary Figure 1 and Supplementary Figure 2). The results showed that the elution contains all the three proteins, indicating a complex of Spr0693–0694–0695 in vitro (Fig. 3a). Notably, only a small fraction of the input prey protein could be pulled down, suggesting the complex is relatively unstable, which is in agreement with the heterogeneous profiles of size-exclusion chromatography (Supplementary Figure 3). The heterogeneity also makes it

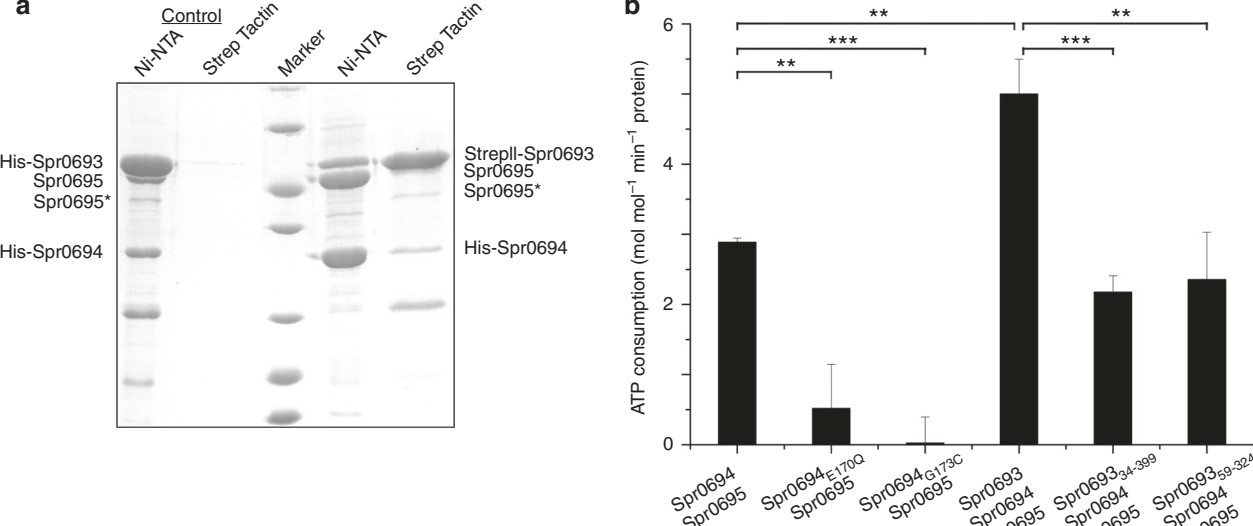

**Fig. 3** Spr0693 interacts with Spr0694–0695 and augments the ATPase activity. **a** Pull-down assays. His-tagged Spr0694 and Strep-tag II tagged Spr0693 were mixed and subjected to Ni-NTA followed by Strep Tactin purification (last two lanes). The first two lanes represent the negative control with His-tagged Spr0694 and His-tagged Spr0693. The eluates of Ni-NTA and Strep Tactin were shown on the PAGE. Spr0695* represents a partially degraded fragment of Spr0695, as confirmed by mass spectrometry (Supplementary Figure 2). The protein marker presented seven bands from bottom to top (14.4, 18.4, 25, 35, 45, 66.2, 116 kDa). **b** The ATPase activities of Spr0694–0695, Spr0693–0694-0695, and mutants. Activity assays were performed in the proteoliposomes. For each reaction, 0.2 μM Spr0694–0695 (or mutants) dimer and/or 0.4 μM Spr0693 (or mutants) hexamer were added to the 100 μL reaction mix. The $V_{max}$ values were calculated by the Pi produced in mole with the consumption of ATP by per mole of Spr0694–0695 protein per minute. At least three independent assays were performed to calculate the means and standard deviations, and the data are presented as the means ± S.D. Two-tailed Student's *t* test is used for the comparison of statistical significance. The *p* values of <0.05, 0.01 and 0.001 are indicated with *, ** and ***, respectively

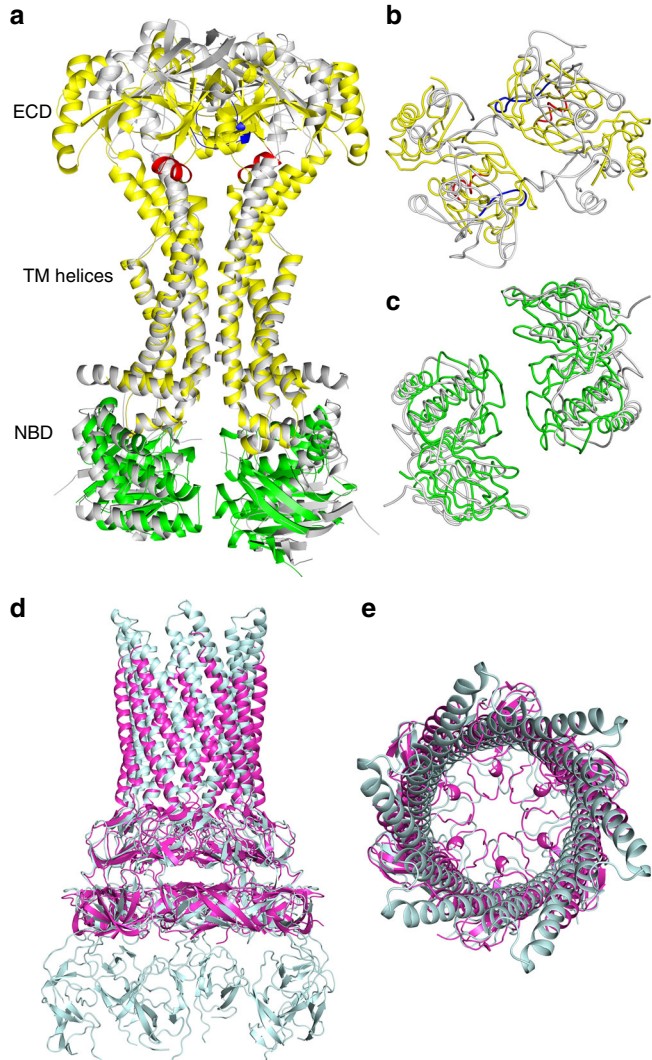

**Fig. 4** Superposition of Spr0694–0695 and Spr0693 to MacB and MacA, respectively. **a** Comparison of the overall structures of Spr0694–0695 and MacB, by superimposing the TM helices in the intracellular membrane face. Spr0694 and Spr0695 are shown in green and yellow, respectively, whereas MacB is shown in gray. The helix α6 and its counterpart in MacB are colored in red and blue, respectively. **b** Top and **c** bottom view of **a**. **d** Side and **e** top view of the superimposed Spr0693 (purple) and MacA (cyan) hexamer

difficult to obtain the diffraction-quality crystal of the full assembly of Spr0693–0694–0695.

The ATPase activity assay is a classic method for evaluating the functionality of ABC transporters[26]. As expected, Spr0694–0695 possesses the ATPase activity, further confirming that it is an ABC transporter. In addition, an E170Q mutation of Spr0694 drastically reduced the ATPase activity, indicating the critical role of catalytic residue Glu170. Previous studies on ABC transporters revealed that the binding and hydrolysis of ATP at the dimeric interface of NBDs trigger conformational changes, which are required for substrate transport[1]. Structural analysis revealed that Gly173 of Spr0694 is located at the dimeric interface of NBDs and has a distance of 8 Å with its counterpart residue of the symmetric subunit. To test whether the dimeric NBDs undergo conformational changes upon ATP hydrolysis, we mutated Gly173 of Spr0694 into cysteine to cross-link the two NBD subunits at the oxidative condition. Electrophoresis profiles

clearly confirmed that the G173C mutant of Spr0694 forms an intermolecular disulfide bond in solution in the presence of $Cu^{2+}$ (Supplementary Figure 4), which means that the NBD dimer is covalently fixed in a closed state. Consequent ATPase activity assays showed that this mutation almost completely abolished the activity (Fig. 3b), which confirms the conformational changes are critical for the ATP hydrolysis cycle for this TM8-fold group of ABC transporters.

Furthermore, to test the contribution of Spr0693 to the ATPase activity, we performed the activity assays in the presence of Spr0693 and its mutants. Addition of the full-length Spr0693 significantly augments the ATPase activity of Spr0694–0695 to two folds (Fig. 3b), indicating the essential role of Spr0693 for the full function of the efflux pump, which is also in agreement with that of MacAB[27,28]. By contrast, addition of either the TM-truncated Spr0693 (Spr0693$_{34–399}$) or TM&MP-truncated Spr0693 (Spr0693$_{59–324}$) showed no significant change of ATPase activity (Fig. 3b). The results clearly demonstrated that the TM and MP domain of Spr0693 are also essential for the full function of Spr0693–0694–0695, similar to that of MacAB[27,28].

**Modeling the assembly of Spr0693–0694–0695 efflux pump.** In agreement with a sequence identity of 45% between the Spr0694 and the NBD of MacB, superposition of the two NBD monomers gives a quite small root-mean-square deviation (RMSD) of 1.43 Å over 218 Cα atoms. In fact, the ECD of Spr0695 also shares a similar structure with the PLD of MacB, with an RMSD of 2.92 Å over 164 Cα atoms. However, structural superposition of Spr0694–0695 against *E. coli* MacB in the MacAB-TolC assembly (PDB: 5NIK) reveals only 780 out of 1215 Cα atoms could be aligned due to drastic conformational changes of both NBD and ECD domains. As shown in Fig. 3a, the transmembrane moieties of TM helices could be well superimposed against each other; in contrast, the dimerized ECDs/PLDs and NBDs undergo significant conformational changes, resulted from the rigid-body rotation and shift. These changes are mainly derived from the rotation of the extracellular segments of TM1 and TM2 of Spr0695 against the symmetric axis at an angle of about 22° and 19°, respectively (Fig. 4a).

For a better comparison, the top view and bottom view of Fig. 4a are shown in Fig. 4b, c, respectively. In contrast to a 25 Å wide interdomain crevice between the PLDs of MacB dimer (Fig. 4b), the ECDs of Spr0695 possess a much narrower crevice of 10 Å from the top view, resulted from a rigid-body rotation in combination with an interdomain shift towards each other (Fig. 4b). In consequence, the previously noted short helix α6 of Spr0695-ECD is docked onto the top moieties of TM helices in Spr0695. However, the helix of MacB corresponding to our α6 is partially unfolded and shifts away in a distance of ~13 Å (Fig. 4b). Seen from the bottom of NBDs (Fig. 4c), Spr0694 rotates ~8.0° towards each other and form a buried interface of ~370 Å², compared to the two separated NBD subunits (~13 Å apart) of MacB. Taken together, the conformational changes in NBDs could be transferred to ECDs, and vice versa, across the membrane via a relatively rigid transmembrane moiety, which serves as a lever fulcrum.

Structural superposition reveals that the hexameric Spr0693 shares an overall architecture quite similar to that of MacA, with an RMSD of 2.85 Å over 1226 Cα atoms. The majority of the variation is due to that each α-hairpin pair of Spr0693 undergoes a slight rotation of ~3° from the counterpart of MacA (Fig. 4d). Notably, the helices of the α-helical domain of Spr0693 are shorter than those of MacA, due to the flexibility of residues from Pro142 to Gly163 at the turn of α-hairpin that could not be traced in our electron density map. Seen at the top,

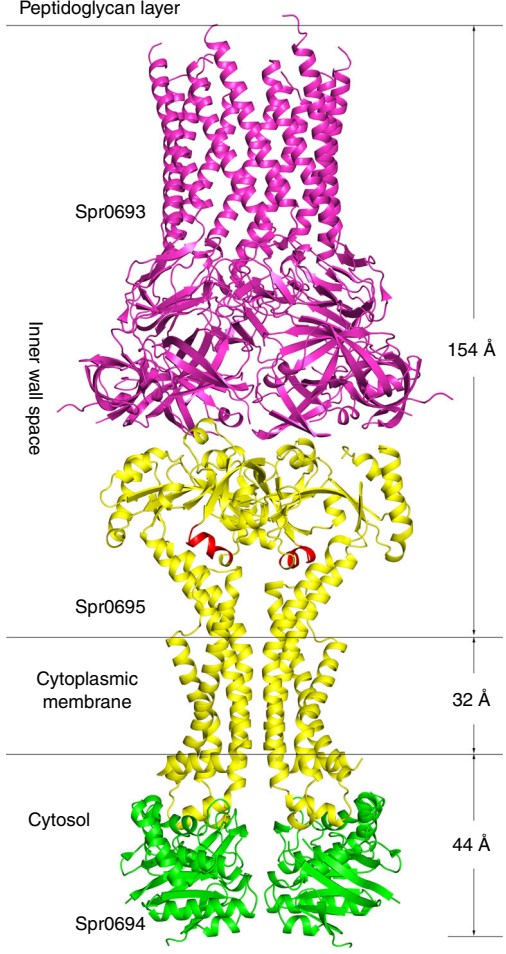

Peptidoglycan layer

Spr0693

Inner wall space

154 Å

Spr0695

Cytoplasmic membrane

32 Å

Cytosol

44 Å

Spr0694

**Fig. 5** The Spr0693-0694-0695 assembly. A simulated model of Spr0693-0694-0695 assembly was manually built by separately superimposing the structures of Spr0693 and Spr0694-0695 to the counterparts of MacAB-TolC assembly (PDB: 5NIK)

Spr0693 and the MacA possess a similar shape of gating ring formed by the loops of the lipoyl domains (Fig. 4e); however, their gating rings differ in constrained residues and the innermost residue (Gly220 in Spr0693 vs. Gln209 in MacA). The residue Gln209 of MacA is conserved in the homologs from Gram-negative bacteria, and these six residues of Gln209 form an inter-protomer hydrogen-bonding network which is thought to be related to the one-way of outward-only transfer[21]. However, Gly220 of Spr0693 is not conserved, and the hydrogen-bonding network is absent in our gating ring. The high B-factor of the gating ring in both structures of Spr0693 and MacA suggest that these loops are very flexible and should undergo drastic conformational changes upon substrate access.

Structure comparisons combined with pull-down and ATPase activity assays strongly suggest that Spr0693–0694–0695 might adopt an assembly mode similar to MacAB. To gain insights into the complex assembly, we manually built a model of Spr0693–0694–0695 complex structure (Fig. 5), by superimposing the structures Spr0693 and Spr0694–0695 onto MacA and MacB of MacAB-TolC structure, respectively. Despite the model has some defects due to the lack of TM and MP of Spr0693, the overall structure and interaction pattern of Spr0693 with Spr0694–0695 are reliable. The complex of Spr0693–0694–0695 also shows a stoichiometry of 6:2:2 for the three components, similar to that of MacAB. The hexameric ring of head-to-tail aligned β-barrel domains of Spr0693 covers the two ECDs of

Spr0695, though the interface between Spr0693 and Spr0695 is much smaller compared to that of MacA and MacB due to the truncation of the MP domain in addition to the shift of ECDs downwards in our structure. It is noteworthy that the distance between the apex of the assembly and the extracellular face of the membrane is about 154 Å (Fig. 5), which is comparable to the reported thickness of inner wall space between the plasma membrane and the peptidoglycan layer in Gram-positive bacteria[29]. The nanotube channel may serve as a tunnel to transport the substrate into the peptidoglycan layer, probably required for avoiding direct access of these substrates/toxins to the inner membrane.

**A putative substrate-entrance tunnel**. Given that MacAB and Spr0693–0694–0695 are efflux pumps as deduced from the previously reported physiological functions[14,17,18], there should exist a selective tunnel for the transport of substrate. Applying the two structures to the program CAVER[30] reveals a tunnel in the MacAB that starts at the cleft between the MP domain and the PLD domain, runs through an acidic pocket on TM helices to the central crevice between the two PLDs of TMDs, and finally reaches the MFP channel beneath the gating ring (Fig. 6a). Notably, a segment of much higher B-factor in PLDs of MacB is partially unfolded and shifts away from the TM helices, thus making an open entrance in the periplasmic space (Fig. 6b). However, we failed in simulating a continuous tunnel in the Spr0693–0694–0695 assembly, as superposition against the MacAB structure reveals two significant variations in the model of Spr0693–0694–0695 that might interrupt the putative route. The crevice between the ECDs of Spr0695 is narrowed to 10 Å in width, compared to the 25 Å wide one in MacB. Moreover, the counterpart helix α6 in Spr0694–0695 is docked onto a pocket, formed by the upper moieties of TM helices and the loop between TM3 and TM4 (Fig. 6c), resulting in the blockage of the entrance of the tunnel. Notably, the pocket, which is composed of residues Asp49, Glu281, Glu374, and Glu377, is generally acidic, complementary to the helix α6 consisting of several positively charged residues. Compared to its counterpart in MacB which is flexible (Fig. 6b), analysis of B-factors revealed that helix α6 of Spr0695 adopts a rigid conformation due to extensive interactions with the TM helices (Fig. 6c). Thus, we propose that the helix α6 functions as a guard at the entrance of substrate tunnel, thus terming it the guard helix. The predominantly acidic and partially hydrophobic tunnel further indicated that Spr0693–0694–0695 most likely transports basic and/or hydrophobic substrates, in agreement with the feature of substrates previously identified by physiological experiments[13].

To test the importance of the guard helix to the function of Spr0693–0694–0695, we tried to truncate the guard helix and prepare the protein for ATPase activity assays. However, truncation of the guard helix almost completely disrupted the folding of Spr0695, making it impossible to purify the well folded protein. Instead, we mutated three positively charged residues Lys207, Arg208, and Lys210 on the guard helix of Spr0695 to alanine (termed *spr0693–0694–0695_M3*), and tested the ATPase activity. The in vitro ATPase activity assays revealed a reduced activity upon these mutations, independent of the addition of Spr0693, as compared to the wild-type proteins (Fig. 6d).

A previous report suggested that the antimicrobial peptide LL–37 is a putative substrate of Spr0693–0694-0695[14]. We, thus, performed the antimicrobial susceptibility assays, which revealed a minimum inhibitory concentration (MIC) of 8 µg/mL for the *spr0693–0694-0695* deletion strain, compared to 32 µg/mL of the wild-type *S. pneumoniae* R6 strain (Table 1, Supplementary

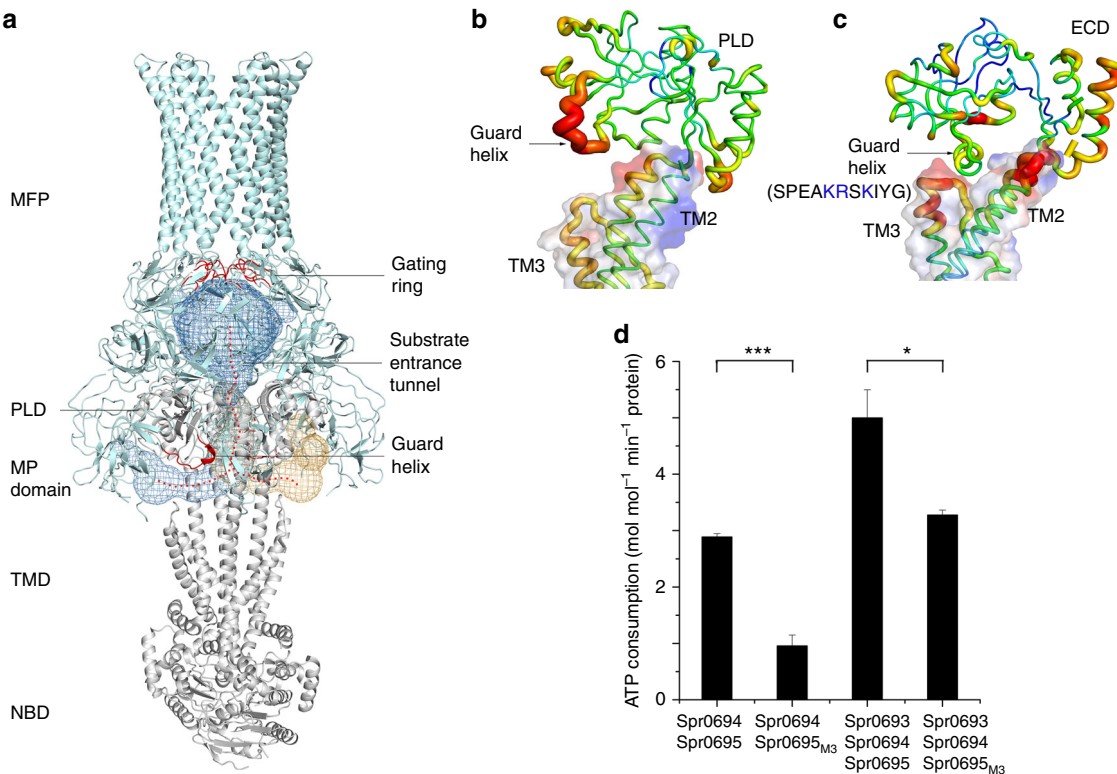

**Fig. 6** A putative substrate-entrance tunnel. **a** A putative substrate-entrance tunnel in the MacAB structure (PDB: 5NIK) calculated by the program CAVER 3.0.1. The tunnel is shown in mesh and the route is indicated as dotted red lines. The gating ring and the guard helix are colored in red. The B-factor putty presentation of (**b**) MacB and (**c**) Spr0695. The electrostatics potential of TM helices is shown in transparent surface. The guard helix of Spr0695 docks to the acidic pocket of TM helices and has a relatively lower B-factor, whereas the guard helix of MacB shifts away from the pocket and the B-factor is much higher. **d** $V_{max}$ values of the ATPase activities of Spr0694–0695, Spr0693–0694–0695 and mutants. Spr0695$_{M3}$ represents a mutant that harboring three positively charged residues Lys207, Arg208, and Lys210 on the guard helix substituted to alanine. At least three independent assays were performed to calculate the means and standard deviations, and the data are presented as the means ± S.D. Two-tailed Student's $t$ test is used for the comparison of statistical significance. The $p$ values of <0.05, 0.01 and 0.001 are indicated with *, ** and ***, respectively

Figure 5), further suggesting that Spr0693–0694-0695 is indeed required for *S. pneumoniae* resistance against LL-37. Complementation of the *spr0693–0694–0695* operon restores the MIC value to a level similar to the wild-type strain. However, complementation with *spr0693–0694-0695$_{M3}$*, which harbors three mutations on the guard helix coding region, into *S. pneumoniae* restores the MIC to 16 μg/mL, suggesting that these mutations on the guard helix partly impair the resistance of *S. pneumoniae* against LL-37. To test the hypothesis that the flexibility of the guard helix is important for the substrate entry, we introduced three disulfide bonds between the guard helix and the TM helices (D49C-R208C, K52C-E205C, and E374C-K207C) based on structural analysis, which is proposed to freeze the guard helix in the oxidative condition. Complementation with the operon harboring these mutations (termed *spr0693–0694-0695$_{3C}$*) results in an MIC of 8 μg/mL, similar to the *spr0693–0694-0695* deletion strain. Altogether, these results clearly suggest that the guard helix of Spr0695 is essential for the full function of Spr0693–0694-0695. As shown in our simulated model (Fig. 5), Spr0694–0695 cooperates with Spr0693 to form a continuous tunnel that transports the substrate from the extracellular membrane face to the peptidoglycan layer.

In addition, we speculate that our Spr0694–0695 structure more likely represents a rest state. In this state, the two NBDs adopt a somewhat closed conformation but yet accessible for ATP; however, the substrate entrance is blocked by the guard helix α6. At this moment, Spr0693 does not associate with Spr0694–0695, considering the relatively unstable complex of

**Table 1 MIC of LL-37 against wild-type *S. pneumoniae* R6 and mutants**

| Strain | MIC (μg mL⁻¹) |
|---|---|
| WT | 32 |
| KO | 8 |
| Complementation | 32 |
| M3 | 16 |
| 3C | 8 |

Spr0693–0694–0695 in solution. Upon approaching of the substrate, the hexameric Spr0693 packs against the Spr0694–0695 complex. The PLDs of TMD rotate outwards and the guard helix shifts away to form a continuous substrate-entrance tunnel. Thus the structure of MacAB-TolC in complex with an unknown molecule should represent a substrate/inhibitor occluded state. Binding and hydrolysis of ATP in NBDs will trigger a series of conformational changes that are required for the substrate transport, which is supported by the ATPase activity assays of Spr0694 G173C mutant (Fig. 3b). Despite the TM helices pack so tightly against each other in both structures of MacAB-TolC and Spr0694–0695, which is hard to form a central channel for substrate transport across membrane, the membrane-buried helical bundle may act as a fulcrum to transmit the conformational changes from NBDs to ECDs, amplified by the upper moieties of TM helices. However, more structures at

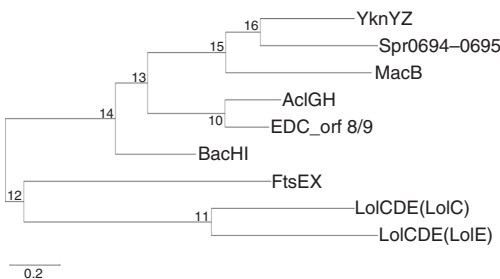

**Fig. 7** Molecular phylogenetic analysis of Spr0694–0695 homologs by Maximum Likelihood method. The sequences of Spr0694–0695 and homologs are from the following organisms (the UniProt code of these proteins is given): *Streptococcus pneumoniae* (Spr0694, Q8DQF8; Spr0695, Q8DQF7; FtsE, A0A064BZ20; FtsX, Q04LE4), *Bacillus subtilis* (strain 168) (YknY, O31711; YknZ, O31712), *Escherichia coli* (strain K12) (MacB, P75831; LolC, P0ADC3; LolE, P75958), *Staphylococcus aureus* (AclG, X4XWR7; AclH, X4Y386), *Staphylococcus epidermidis* (EDC_orf8, H9BG71; EDC_orf9, H9BG72), *Enterococcus faecalis* (BacH, O52970; BacI, O52971)

various states in the transport cycle are needed to figure out the fine transport mechanism.

## Discussion

Besides the structures of Spr0694–Spr0695 and MacB, there are currently 36 unique ABC transporters of known three-dimensional structures (Membrane Proteins of Known 3D Structure, http://blanco.biomol.uci.edu/mpstruc/), 21 of which are exporters. While importers possess 10–20 TM helices, all exporters have 12 TM helices, except for the recently reported 10-TM exporter LptB$_2$FG[31]. Interestingly, the putative exporters Spr0694–0695 and MacB have only 8 TM helices. Sequence analysis indicated that Spr0694–0695 and homologs could be grouped into the prokaryotic ABC3 family. The structures of Spr0694–0695 and MacB revealed a tightly packed dimeric interface of TMs that provides no space for substrate entrance, suggesting that this family of ABC transporters should apply a transport mechanism other than alternating access. In fact, Spr0694–0695 and MacB possess a putative insertion, namely ECD or PLD, between TM1 and TM2; and further structural analyses enabled us to simulate a substrate-entrance tunnel of lateral substrate access at the inner wall space or periplasm. This lateral access of substrate is also applied by two recently reported transporters. The structure of *E. coli* lipopolysaccharide exporter LptB$_2$FG proposed that the substrate is loaded from the periplasmic side of the inner membrane via the lateral gates between the TMs, and then forwarded by the periplasmic domains[31]. Another case is from the human lipid exporter ABCA1, which also lacks a central TM cavity; however, the ECDs provide a putative tunnel for the access of substrate from the lateral side of TMs[7].

Sequence homology search combined with phylogenetic analyses enabled us to reveal some unique features of ABC3 family transporters, which usually possess an 8-TM fold, and an insertion between TM1 and TM2 (Fig. 7). In addition, these transporters are most likely combined with one or two partners to accomplish various functions. In many cases, these unique ABC transporters are linked to a membrane fusion protein to form an intact efflux pump. For example, the *Bacillus subtilis* transporter YknYZ cooperates with YknX to pump out the endotoxin Sporulation-Delaying-Protein[32]. Similarly, *Staphylococcus aureus* AclGH, *Enterococcus faecalis* BacHI and *Staphylococcus epidermidis* EDC_orf8/9 are all putative transporters of efflux pump that secret different bacteriocins[8,9,33]. Nevertheless, in some other cases, the transporters may cooperate with a partner that is

structurally distinct from MFPs, executing a function other than substrate transport. For example, the *E. coli* LolCDE, cooperating with LolA, catalyzes the release of lipoproteins from the inner membrane to the periplasm[34]. In fact, the *S. pneumoniae* genome encodes a couple of similar protein complexes including the cell division related complex FtsEX, which interacts with the peptidoglycan hydrolase PcsB and regulates the peptidoglycan remodeling[35].

In summary, the structure of Spr0694-Spr0695 reveals some features for this family of ABC transporters, which are characterized by a unique 8-TM fold and an ECD/PLD insertion between TM1 and TM2. Most members in this family usually apply a lateral substrate access and a non-canonical transport mechanism, cooperating with other partner(s) to complete a long distance transport of substrate. More structural and functional investigations of this ABC family will broaden our understanding on the transport mechnism of ABC transporters.

## Methods

**Protein preparation.** The coding region of full-length *spr0694–0695* was amplified from *Streptococcus pneumoniae* R6 genomic DNA by polymerase chain reaction (PCR) with primer pBAD-Spr0694–0695-F1/R2 (Supplementary Table 1) The coding region was cloned into a pBAD-derived vector using One Step Cloning Kit (Vazyme), with an N-terminal 6 × His-tag on Spr0694. The recombinant plasmid was transformed into *E. coli* TOP10 strain (Invitrogen), growing at 37 °C in Luria-Bertani (LB) culture medium (10 g NaCl, 10 g Bacto-Tryptone and 5 g yeast extract per liter), supplemented with 50 μg mL$^{-1}$ ampicillin. Overexpression of Spr0694–0695 was induced by adding 0.02% L-arabinose (Sangon) when the cell density reached OD$_{600}$ nm of 0.6–0.8. The *E. coli* cells were grown in 1 L of LB medium in a shaker (New Brunswick Scientific). After growth for 4 hr at 37 °C, the cells were collected, homogenized in the buffer containing 25 mM HEPES-NaOH pH 8.0, 1 M NaCl, and 5% glycerol and stored at −80 °C before use. For purification, cells were thawed by disrupting with a Constant Cell Disruption System (Constant Systems) with two passes at 35 kpsi. Cell debris was removed by low-speed centrifugation for 10 min. The supernatant was collected and ultra-centrifuged at 200,000 × *g* for 1 h. The membrane fraction was collected and incubated for 1.5 h with the buffer containing 25 mM HEPES-NaOH pH 8.0, 1 M NaCl, 5% glycerol, 1% (w/v) dodecyl-β-D-maltopyranoside (DDM; Anatrace) and 1% (w/v) Lauryl Maltose Neopentyl Glycol (LMNG; Anatrace) at 4 °C. After another round of ultracentrifugation at 200,000 × *g* for 30 min, the supernatant was collected and loaded onto nickel affinity resin (Ni-NTA; Qiagen). The column was then washed with 25 mM HEPES-NaOH pH 8.0, 1 M NaCl, and 5% glycerol, 0.02% DDM and 0.02% LMNG. Contaminants were removed by washing with 25 mM HEPES-NaOH pH 8.0, 1 M NaCl, 5% glycerol, 0.01% DDM and 0.02% LMNG plus 20 mM imidazole. The protein was eluted from the affinity resin with buffer containing 25 mM HEPES-NaOH pH 8.0, 1 M NaCl, 5% glycerol, 0.01% LMNG, supplemented with 400 mM imidazole. Afterwards, the protein was concentrated to 1 mL before further purification by gel filtration (Superdex 200 increase 10/300; GE Healthcare) in the buffer containing 25 mM HEPES-NaOH pH 8.0, 150 mM NaCl, 5% glycerol and 0.01% LMNG. The peak fractions were collected for further procedures.

The full-length coding region of *spr0693* was amplified and cloned into a pET29b vector with a C-terminal 6 × His-tag or Strep-tag II in the same way as *spr0694–0695*. The recombinant plasmid was transformed into *E. coli* BL21 (DE3) cells (Novagen) and grown in LB broth with 30 μg/mL kanamycin. When OD$_{600}$ nm reached 0.6–0.8, cells were induced with 0.2 mM isopropyl-β-D-thiogalactoside (IPTG) at 16 °C for 20 h before collecting by centrifugation at 8,000 × *g* for 5 min. Collected cells were resuspended and stored at −80 °C for further use. Full-length Spr0693 with a His-tag was purified in the same way as Spr0694–0695. In brief, cells were disrupted and cell debris were removed by centrifugation. Cell membrane was collected by ultracentrifuge from the supernatant and was then solubilized in buffer containing 1% LMNG and 1% DDM. The supernatant was then loaded to Ni-NTA column after the insoluble pellets were remove by another round of ultracentrifuge. The proteins were then washed and eluted with buffer contain imidazole and DDM/LMNG. The eluted proteins were further purified by gel filtration in the buffer containing 25 mM HEPES-NaOH pH 8.0, 150 mM NaCl, 5% glycerol and 0.01% LMNG.

To obtain Spr0693 crystals of diffraction quality, we tried a series of truncations, including Spr0693$_{34–399}$, Spr0693$_{59–324}$ and Spr0693$_{70–237}$, based on structure prediction by Phyre2[36]. These plasmids were constructed based on the full-length *spr0693* plasmid and were transformed and expressed in the same way. For purification, cells were thawed and lysed by ultra-sonication, and then centrifuged at 30000 × *g* for 30 min to remove cell pellet. The supernatant was incubated with Ni-NTA column for 1 h and was washed by 20 mL buffer containing 25 mM HEPES-NaOH pH 8.0, 150 mM NaCl, and 5% glycerol, supplemented with 20 mM imidazole. The proteins were eluted with 400 mM imidazole in buffer and desalted

## Table 2 Data collection and refinement statistics

| | Spr0694–0695 (PDB 5XU1) | Spr0693 (PDB 5XU0) |
|---|---|---|
| **Data collection** | | |
| Space group | $C222_1$ | $P4_12_12$ |
| Cell dimensions | | |
| $a, b, c$ (Å) | 141.05, 154.99, 205.89 | 159.21, 159.21, 99.40 |
| $\alpha, \beta, \gamma$ (°) | 90.00 | 90.00 |
| Resolution (Å) | 50.0–3.3 (3.4–3.3)[a] | 50.00–3.0 (3.1–3.0) |
| $R_{merge}$ | 18.5 (85.8) | 9.6 (67.9) |
| $I/\sigma(I)$ | 13.3 (3.2) | 15.6 (1.6) |
| Completeness (%) | 100 (99.9) | 99.8 (98.8) |
| Redundancy | 13.5 (14.0) | 7.0 (7.0) |
| **Refinement** | | |
| Resolution (Å) | 50.0–3.3 | 50.0–3.0 |
| No. reflections | 34,560 | 23,899 |
| $R_{work}/R_{free}$ | 23.4/28.9 | 24.8/30.5 |
| No. of atoms | | |
| Protein | 9,269 | 5,088 |
| Ligand/ion | 2 | 0 |
| Water | 16 | 0 |
| B-factors | | |
| Protein | 84.9 | 101.0 |
| Ligand/ion | 41.5 | |
| Water | 61.2 | |
| R.m.s. deviations | | |
| Bond lengths (Å) | 0.008 | 0.004 |
| Bond angles (°) | 1.298 | 0.793 |

[a] Values in parentheses are for highest-resolution shell
1 crystal was used in the structure of PDB 5XU1; 1 crystal was used in the structure of PDB 5XU0

into the low salt buffer containing 25 mM HEPES-NaOH pH 8.0, 20 mM NaCl. Afterwards, they were loaded onto a QHP anion exchange column (GE healthcare) and were eluted by Hepes buffer containing linear salt concentration from 20 mM to 1 M NaCl. The eluted proteins were concentrated to 5 mL and were further purified by size-exclusion chromatography (Superdex 200 16/60, GE healthcare) equilibrated with 25 mM HEPES-NaOH pH 8.0, 150 mM NaCl, and 5% glycerol and the peak fractions were collected for further procedures.

All mutants were generated with a standard PCR-based strategy (primers listed in Supplementary Table 1) and were cloned, overexpressed and purified the same way as the wild-type protein.

Selenomethionine (Se-Met) labeled proteins were expressed in methionine synthesis deficient strains B834 (DE3) (Novagen) following the standard procedure. Cells transformed with the recombinant plasmid were inoculated into LB medium at 37 °C overnight. The cells were collected and washed twice with the M9 medium. Then cells were grown in the minimal medium (1 g NH$_4$Cl, 3 g KH$_2$PO$_4$, 17 g Na$_2$HPO$_4$, 1% glucose, 2 mM MgSO$_4$, 0.1 mM CaCl$_2$, 1 mg vitamin B1, 7.5 mg FeSO$_4$ per liter) culture with additional 50 mg/L essential amino acid (Se-Met instead of methionine) and 30 µg/mL kanamycin. When OD$_{600}$ nm reached 0.6–0.8, cells were induced with 0.2 mM IPTG at 16 °C for 20 h before collecting. Collected cells were resuspended in the lysis buffer containing 25 mM HEPES-NaOH pH 8.0, 150 mM NaCl, and 5% glycerol and stored at −80 °C. Se-Met labeled proteins were purified the same as native proteins.

**Crystallization.** The purified Spr0694–0695 protein was concentrated by ultra-filtration (Millipore Amicon) to 12 mg/mL and incubated with 10 mM MgCl$_2$ and 2 mM AMP-PNP for 2 h on ice before crystallization. Initial crystal screening was performed using sitting-drop vapor diffusion method with MemGold, MemSys, MemStart and MemPlus crystallization kits (Molecular Dimensions). Crystals appeared in the next day in 0.2 M ammonium sulfate, 0.1 M sodium acetate 4.6, 28% v/v PEG 550 MME and grew to its maximum size in 3–4 days. After optimization with Additive Screen kit (Hampton Research), we got better single crystals with the additive of cadmium chloride. The optimized crystallization solution for data collection is 0.2 M ammonium sulfate, 0.1 M sodium acetate 5.0, 25–31% v/v PEG 550 MME and 0.01 M cadmium chloride.

The purified Spr0693$_{34–399}$, Spr0693$_{59–324}$, Spr0693$_{70–237}$ proteins were concentrated to 5 mg/mL for crystallization. All of these constructs yielded crystals, but only the crystals of Spr0693$_{59–324}$ could be optimized to a diffraction at 3 Å. The Spr0693$_{59–324}$ crystals were obtained at 12 °C by hanging-drop vapor diffusion using a reservoir solution containing 100 mM sodium acetate pH 4.5, 7.5% PEG 4000, 100 mM CaCl$_2$, 10%–20% MPD. Dehydration was performed by adding

MPD to the reservoir solution to a final concentration of 25% 48 h before crystals collection. In order to gain strong anomalous signal of Se-Met-labeled crystals, T197M/I317M double mutation was introduced to Spr0693$_{59–324}$, and the Se-Met crystals were obtained in a similar condition.

**Data collection and processing.** Both native data of Spr0694–0695 and Se-Met-derivative data of Spr0693$_{59–324}$ were collected from a single crystal at 100 K in a liquid nitrogen stream on beamline BL19U[37] at the Shanghai Synchrotron Radiation Facility (SSRF) of China at the wavelength of 0.97775 Å. The data set of Spr0694–0695 was processed using HKL2000[38] and the data set of Spr0693$_{59–324}$ was processed with XDS[39,40]. Crystals of Spr0694–0695 and Spr0693$_{59–324}$ belong to $C222_1$ and $P4_12_12$, respectively (Table 2).

**Structure determination and model building.** The structure of Spr0694–0695 was solved by molecular replacement with Molrep in CCP4[41] using crystal structure of MJ0796 ATP-binding cassette (PDB: 1F3O) as a search model. Two NBDs were found in the initial model. After a few cycles of manual building in Coot[42] and refinement in REFMAC implemented in CCP4, electron density map showed clear features of the TM helices, which enabled us to manually build the TM helices in Coot. After a few cycles of refinement, the density is greatly improved and additional map was found to build ECDs. We, thus, automatically built the ECDs by BUCCANEER implemented in CCP4. As the model was almost completed, the side-chains of residues on NBDs and ECDs were manually assigned in Coot with the automatically building by BUCCANEER. The model was then checked in Coot and refined with REFMAC and Phenix.refine[43].

The crystal structure of Spr0693$_{59–324}$ was determined by the single-wavelength anomalous dispersion (SAD) phasing technique from a single Se-Met-substituted protein crystal to a maximum resolution of 3.0 Å. The AutoSol program from PHENIX[43] was used to locate the Se atoms and to calculate the initial phases, yielding FOM and BAYES-CC values of 0.292 and 57.73, respectively. Each asymmetric unit contains three Spr0693 molecules. The initial electron-density maps showed clear features of secondary structural elements, permitting the building of approximately half of the structure in PHENIX, yielding a model-CC of 0.61. The initial models were then improved with BUCCANEER in CCP4 and Phenix.autobuild[43]. Afterwards, the models were improved by cycles of manual building in Coot and refinement in REFMAC and PHENIX.

The models of Spr0693 and Spr0694–0695 were assessed with MolProbity[44], which indicated 99% of residues in the allowed regions of the Ramachandran plot for both models (Supplementary Figure 6). All structural figures were generated with Pymol[45].

**Pull-down assays.** Cells expressing N-terminal His-tagged Spr0694–0695 and cells expressing C-terminal Strep-tag II tagged Spr0693 were mixed and then disrupted by ultra-sonication. For the negative controls, cells expressing His-tagged Spr0693 were used. After extracting from membrane, the proteins were first loaded to a Ni-NTA column as mentioned above and eluted with 250 mM imidazole. The eluate was then incubated with Strep Tactin resin (Qiagen) in 4 °C for 1 h. After washing several times with 25 mM Na$_3$PO$_4$ pH 8.0, 300 mM NaCl, 0.02% DDM, the proteins were eluted with 2.5 mM desthiobiotin (Sigma). All samples were applied to sodium dodecyl sulfate polyacrylamide gel electrophoresis and staining. The corresponding bands were further identified by mass spectrometry.

**ATPase activity assays.** All ATP hydrolysis assays were performed in the proteoliposomes. E. coli polar lipids (Avanti) were resuspended in 20 mM HEPES-KOH (pH 7.0) to the final concentration of 20 mg mL$^{-1}$. A final concentration 0.45% of Triton X-100 was added to destabilize the liposomes for 0.5 h at room temperature. The proteins were added to the destabilized liposomes and incubated for 1 hr at 4 °C. The ratio of lipids and proteins was kept at 100:1 (w/w). Triton X-100 was removed by SM-2 Adsorbent Bio-Beads (Bio-Rad) by incubation at 4 °C overnight and repeated for another 2 h. Proteoliposomes were diluted and resuspended twice with the ice cold buffer by 250,000 × $g$ for 1 h at 4 °C. After the final centrifugation, the pellets were resuspended in 20 mM HEPES-KOH, pH 7.0, 50 mM KCl and 2 mM MgCl$_2$ to 100 µL as one reaction sample. For each reaction sample, 0.2 µM of Spr0694–0695 dimer and 0.4 µM of Spr0693 hexamer or mutated proteins were added.

ATP was added to each sample at a final concentration of 2 mM by three rounds of freezing in liquid nitrogen followed by thawing in a bath sonicator. Reactions were performed at 37 °C for 30 min and the amount of released Pi was quantitatively measured using the ATPase colorimetric Assay Kit (Innova Biosciences) in 96-well plates at OD$_{650}$ mm.

**Bacterial strains and growth conditions.** All S. pneumoniae strains used in this study are listed in the Supplementary Table 1 and were stored in 15% glycerol at −80 °C. The strain R6 (ATCC BAA-255)[46] was used as the parental strain. The strains were cultured in Todd-Hewitt broth (BD Biosciences) supplemented with 0.5% yeast extract (termed THY) or on tryptic soy agar (Difco) plates containing 5% (v/v) sheep blood. When necessary, kanamycin (400 µg/mL) or streptomycin (150 µg/mL) or tetracycline (1 µg/mL) was included in the broth and agar media

for selection purpose. All *S. pneumoniae* strains were cultured at 37 °C with 5% $CO_2$.

### Construction of *S. pneumoniae* strains and plasmids

The chromosomal *spr0693–0694-0695* deletion mutant of *S. pneumoniae* was generated from the strain R6 by allelic replacement as described previously[47]. In brief, the 862-bp upstream region of *spr0693* and 1008-bp downstream regions of *spr0695* in strain R6 were separately amplified from genomic DNA by PCR with primer pairs Up-F/R and Down-F/R (Supplementary Table 1), respectively. The kanamycin-resistant Janus cassette fragment was amplified from the prepared genomic DNA of *S. pneumoniae* strain ST588[48] using primers JC-F/R (Supplementary Table 1). These fragments were integrated by overlapping PCR. Purified PCR products were then transformed into the strain R6 to select kanamycin-resistant colonies on blood agar plates. The replacement of the whole *spr0693–0694-0695* coding region with the Janus cassette was selected by kanamycin and was then detected and confirmed by PCR and DNA sequencing.

A $Zn^{2+}$ inducible pJWV25 plasmid[49] was used for complementation. The coding region of full-length *spr0693–0694-0695* was cloned into pJWV25 using One Step Cloning Kit (Vazyme), with N-terminal Protein A-tag and 3xFlag-tag on Spr0693. To further investigate the function of the guard helix, we constructed another two mutants, K207A-R208A-K210A and D49C-R208C/K52C-E205C/E374C-K207C, designated as pJWV25-M3 and pJWV25-3C, respectively (Supplementary Table 1). These plasmids were transformed into the *spr0693–0694-0695* deletion strain similar to the knock-out PCR products, and the strain was then grown on blood agar plates supplemented with 1 µg/mL tetracycline. Colonies harboring the target plasmids were further confirmed by PCR.

### Antimicrobial susceptibility assays

The Broth microdilution MIC assays were performed in 96-well microtiter plates following the NCCLS M7 procedure[50]. The five different strains, WT, KO, Complementation, M3, 3C, were thawed and resuspended in fresh THY medium supplemented with 0.3 mM $ZnCl_2$ for induction and were then cultured at 37 °C with 5% $CO_2$. When $OD_{620}$ nm reached 0.5, the bacteria were 1:100 diluted to fresh THY supplied with $ZnCl_2$ and were then added into the 96-well plate at a total volume of 200 µL per well. After preliminary screening, the final concentrations of LL-37 were diluted into a gradient of 0, 1, 2, 4, 8, 16, 32, 64 µg/mL. The microtiter plates were then incubated at 37 °C in a microplate reader (CLARIOstar, BMG LABTECH) for overnight growth, and the $OD_{620}$ nm was measured every 30 min. MICs were calculated from the lowest concentration of LL-37 that was able to inhibit the growth of the tested strains.

### Phylogenetic tree building

The evolutionary history of Spr0694–0695 and homologs is inferred by using the Maximum Likelihood method based on the Poisson correction model[51]. The tree with the highest log likelihood (−6371.51) is shown. Initial tree(s) for the heuristic search were obtained automatically by applying Neighbor-Join and BioNJ algorithms to a matrix of pairwise distances estimated using a JTT model, and then selecting the topology with superior log likelihood value. The tree is drawn to scale, with branch lengths measured in the number of substitutions per site. The analysis involved in 9 sequences. All the positions containing gaps and the missing data were eliminated. There are a total of 288 positions in the final dataset. Evolutionary analyses are conducted in MEGA7[52].

### Data availability

The atomic coordinates and structure factors of Spr0694–0695 and Spr0693 are available at the Protein Data Bank with the access code 5XU1 and 5XU0, respectively. Other relevant data can be obtained from the corresponding authors on reasonable request.

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

## Acknowledgements

This work is supported by the Ministry of Science and Technology of China (http://www.most.gov.cn, Grants No. 2015CB910100, 2014CB910100 and 2016YFA0400900), the National Natural Science Foundation of China (http://www.nsfc.gov.cn, Grant No. 31600599, 31400628 and 31621002), the Strategic Priority Research Program of Chinese Academy of Sciences (http://www.cas.cn/, Grant No. XDB08020300), and the Innovation Center for Cell Signaling Network. We thank the staff at the Shanghai Synchrotron Radiation Facility and the Core Facility Center for Life Sciences at University of Science and Technology of China for technical assistance.

## Author contributions

C.-Z.Z. and Y.C.: Conceived, designed and supervised the project. C.-Z.Z., Y.C., Y.-L.J., W.-T.H. and H.-B.Y.: Analyzed the data and wrote the manuscript. H.-B.Y., M.-T.C., and W.-T.H.: Carried out the protein purification and crystallization. H.-B.Y. and M.-T.C.: Performed crystal screening and optimization. Y.-L.J. and H.-B.Y.: Performed the data collection and structural determination. All authors discussed the data and read the manuscript.

## Additional information

**Competing interests:** The authors declare no competing financial interests.

**11**