## [Peer Review File · Nature Communications]

Editorial Note: This manuscript has been previously reviewed at another journal that is not operating a transparent peer review scheme. This document only contains reviewer comments and rebuttal letters for versions considered at Nature Communications. Mentions of prior referee reports have been redacted.

Reviewers' comments:

Reviewer #1 (Remarks to the Author):

The paper reports crystal structures of the ABC transporter and its accessory protein from a Gram-positive bacterium *S. pneumoniae*. Most of the studies of these transporters are done on the Gram-negative homolog MacAB from *E. coli* or other Gram-negative bacteria. This is the first structural insight into a Gram-positive representative. The study is interesting and will stimulate further structural and functional studies of these transporters. The major findings are that Spr proteins are indeed very much MacAB-like. The transporter is very similar to MacB but in a different conformation from the one recently reported in fully assembled complex MacAB-TolC using cryo-EM. The truncated structure of Spr0693 is very similar to MacA but lacks the membrane proximal and the transmembrane domains. There are a few weaknesses:

1. The manuscript is written, as if nothing is known about these proteins and their differences from other ABC importers and exporters. The four TMS topology of MacB and YknYZ is known for at least ten years. Their biochemical properties and activities are characterized. The introduction shows that the authors are not very familiar with these proteins and the field. These transporters belong to the ABC3 family, subfamily "Macrolide Exporters" 3.A.1.122. The Intro should be focused on what is known about closest Spr homologs and what is unknown and exciting about these proteins.
2. All mechanistic conclusions are purely speculative and do not have any experimental support. The model is built based on the cryo-EM structure of MacAB-TolC, which is reasonable given a significant similarity. But Spr0693, as well as MacA and YknX, all contain a single transmembrane domain. This domain is essential for MacA function. Hence, each hexameric Spr0693 or MacA contributes six transmembrane domains that encircle the transmembrane domain of Spr and MacB transporters, interact with their NBDs and stimulate activities of complexes. The model based on MacAB-TolC could be misleading.

Reviewer #2 (Remarks to the Author):

Yang et al report the crystal structures of spr0693 and spr0694/0695, a MacAB like system from the Gram-positive bacterium *S. pneumoniae*. Based on the available cryo-EM structure of MacAB from *E. coli*, the authors construct a model of the entire complex. Based on a comparison with the MacAB complex structure, the authors conclude that their structure corresponds to the resting state of the system, while MacAB represents the occluded state. In my opinion this is a very important manuscript as it provides the first piece of information on a MacAB-like system from Gram-positive bacteria. Furthermore, a detailed comparison of the MacAB system and their structures revealed commonalities and differences between both systems that allowed the authors to propose a model of how this resistance systems work. Finally, the TMD of spr0694/0695 contains only four TMHs and adopts a novel fold among ABC transporters. In my opinion, the manuscript deserves publication after some minor issues have been clarified.

To obtain suitable crystals of spr0693, truncations were performed. What is the impact of these truncation of the functionality of the MacAB-like system? Is it still a resistance machinery? In my

opinion this is an important point as the model is based on the truncated form of spr0693 and also used to propose a 'mode of action'

The identified guard helix (helix 6) should be supported by a rational mutation that destroys the interaction. What is the functional consequence of such a mutation?

Line 35 / 36: ABC import systems have been described in plants as well.

Line 298: Protein elution was indeed performed in the absence of any imidazole?

Lines 310- 321: More details must be provided, otherwise no one could follow the details of the purification procedure

Figure 4a: A high quality gel is required as it nearly impossible to grasp anything from the current gel.

Reviewer #3 (Remarks to the Author):

The authors describe a structure of the (Gram-negative) MacAB homologue from a Gram-positive bacterium, encoded by the gene cluster spr0693-0694-0695. The structure is derived from crystallization and using SeMet-phasing for the MacA homolog Spr0693 and using molecular replacement with the MJ0796 ATP-binding cassette protein for Spr0694-0695. The manuscript describes the structure of the Gram-positive MacAB homologue and the comparison with the E. coli MacAB structure and apart from a (not so convincing) pull down experiment, no additional experiments have been made to address the fascinating questions which arise from the structure.

For the manuscript to be valuable to the reader, some of the following questions have to be addressed to make the manuscript more substantial for possible publication in Nature Communications in this referees opinion. At the moment I would not recommend publication:

1. The activity of the ATPase should be addressed. Experiments in proteoliposomes must show activity of the Spr0693-0694-0695 complex. For MacAB it has been shown that MacA stimulates ATPase activity of MacB manifold. This observation for the Spr0693-0694-0695 complex would strengthen the necessity for complex formation of a MacAB homologue from a Gram-positive bacterium
2. The guard helix appears to be an interesting and important feature as observed from the structure. However, it is not clear what the helix is guarding. From Fig. 5a it is not evident that it blocks the lateral entrance. Since the guard helix is rich in positive side chains, as the putative substrate LL-37, it would be very helpful to have some structure/function activity relationship experiments, including variants with substituted side chains in the guard helix.
3. The manuscript is also in need for a morphing (movie) between the postulated resting state of the Spr0694-0695 complex and the postulated occluded state of MacB.
4. The complex formation data are not very convincing. After Ni-NTA purification, assuming only spr0694-0695 His-tagged protein in the solubilized material, how does the SDS-PAGE analysis look like? The gel shown only shows the result after Ni-NTA and StrepII-tag pull-down. Is the complex stable in e.g. SEC? The stained gel in Fig. 4a shows also shows a small amount of protein loaded. The pull-down needs to contain proper controls, like a pull-down with non-tagged complex and a positive control with e.g. MacAB from E. coli. Data could be substantially supported by MS.
5. In Fig 4a, one of the bands is designated to a truncated version of Spr0695. How do the authors reach that conclusion? Mass-spec data would help to verify the identity of the band in the gel.
6. Is the MacA homologue Spr0693 capable to interact with TolC in E. coli. Does expression of the Spr0693-0694-0695 complex in E. coli decrease susceptibility against e.g. Erythromycin, fusidic acid, chloramphenicol?
7. Is it possible to express the Spr0693-0694-0695 complex in its cognate host? If yes, a pull down of

this complex might reveal other proteins attached. This could reveal more on the functional role, since it is puzzling why spr0694-0695 is expressed in an operon with spr0693. The reason to have such a complex in Gram-positive bacteria is one of the main questions.

8. The change in the conformation of the NBD domains between the resting and occluded states could be analyzed via Cys-crosslinking (single Cys variants have to be constructed). It verifies the structure and gives insight in the conformational changes during the ATP hydrolysis cycle, which is still elusive for this TM8-fold group of transporters compared to the other ABC transporters.

9. The flexible N211-V224 loop in Spr0693 is said to be non-conserved and this would be an indication for a rather broad spectrum of substrate selectivity. I cannot follow this line of argumentation. The other hypothesis for this loop to be a valve preventing backflow, is already difficult to follow in a MacAB-ToIC setup, but in a MacAB like setup as suggested for the Spr0693-0694-0695 complex seems not to make much sense, since the substrates are exported in the medium anyway, whether they go through spr0693 or are only transported by Spr0694-0695 and end up in the outside medium as well.

10. I wonder if the docking of Spr0693 to Spr0694-0695 makes much sense, since the ECDs have a different conformation. Is it anyway likely that Spr0693 and Spr0694-0695 are forming a complex at all times?

11. Is Spr0693 attached to the membrane? Is complex formation and disassembly of the complex thought to be transient?

12. Do the authors have a suggestion why a Gram-positive organism needs to have a bipartite setup including a nanotube channel protein?

The last question, the role of the guard helix, and the stability of the bipartite complex should be supported with functional data to underpin the structural information the authors obtained from the structure. The structure alone is interesting, but we do not learn very much on the structure/function relationship of this group of transporters within the ABC transporter family.

Reviewers' comments: Reviewer #1 (Remarks to the Author):

The paper reports crystal structures of the ABC transporter and its accessory protein from a Gram-positive bacterium *S. pneumoniae*. Most of the studies of these transporters are done on the Gram-negative homolog MacAB from *E.coli* or other Gram-negative bacteria. This is the first structural insight into a Gram-positive representative. The study is interesting and will stimulate further structural and functional studies of these transporters. The major findings are that Spr proteins are indeed very much MacAB-like. The transporter is very similar to MacB but in a different conformation from the one recently reported in fully assembled complex MacAB-TolC using cryo-EM. The truncated structure of Spr0693 is very similar to MacA but lacks the membrane proximal and the transmembrane domains. There are a few weaknesses:

1. The manuscript is written, as if nothing is known about these proteins and their differences from other ABC importers and exporters. The four TMS topology of MacB and YknYZ is known for at least ten years. Their biochemical properties and activities are characterized. The introduction shows that the authors are not very familiar with these proteins and the field. These transporters belong to the ABC3 family, subfamily "Macrolide Exporters" 3.A.1.122. The Intro should be focused on what is known about closest Spr homologs and what is unknown and exciting about these proteins.

A: Sorry for the misleading. We have added more information in the Introduction part to describe the properties of these transporters that belong to the ABC3 family. Accordingly, we have revised the corresponding statements.

2. All mechanistic conclusions are purely speculative and do not have any experimental support. The model is built based on the cryo-EM structure of MacAB-TolC, which is reasonable given a significant similarity. But Spr0693, as well as MacA and YknX, all contain a single transmembrane domain. This domain is essential for MacA function. Hence, each hexameric Spr0693 or MacA contributes six transmembrane domains that encircle the transmembrane domain of Spr and MacB transporters, interact with their NBDs and stimulate activities of complexes. The model based on MacAB-TolC could be misleading.

A: According to your suggestion, we performed a series of ATPase activity assays to verify the contribution of Spr0693 to the ATPase activity (Fig. 3b). Addition of the full-length Spr0693 significantly augments the ATPase activity of Spr0694-0695 to 2 folds. By contrast, addition of the TM-truncated Spr0693 (Spr0693₃₄₋₃₉₉) to Spr0694-0695 showed no significant change of the ATPase activity. The results clearly demonstrated that the TM of Spr0693 is essential for the full function of Spr0693-0694-0695, consistent with the previous report of MacAB¹. As you proposed, the real complex of MacAB, as well as Spr0693-0694-0695, most likely adopts an overall architecture with the six transmembrane helices encircling the transmembrane domain of Spr0694-0695 or MacB. However, due to the deletion of MP domain and transmembrane helix in our Spr0693 structure, and the invisible transmembrane helices in the Cryo-EM structure of MacAB, it is hard to construct a model with the full-length Spr0693 or MacA.

Reviewer #2 (Remarks to the Author):

Yang et al report the crystal structures of spr0693 and spr0694/0695, a MacAB like system from the Gram-positive bacterium *S. pneumoniae*. Based on the available cryo-EM structure of MacAB from *E. coli*, the authors construct a model of the entire complex. Based on a comparison with the MacAB complex structure, the authors conclude that their structure corresponds to the resting state of the system, while MacAB represents the occluded state. In my opinion this is a very important manuscript as it provides the first piece of information on a MacAB-like system from Gram-positive bacteria. Furthermore, a detailed comparison of the MacAB system and their structures revealed commonalities and differences between both systems that allowed the authors to propose a model of how this resistance systems work. Finally, the TMD of spr0694/0695 contains only four TMHs and adopts a novel fold among ABC transporters. In my opinion, the manuscript deserves publication after some minor issues have been clarified.

1. To obtain suitable crystals of spr0693, truncations were performed. What is the impact of these truncation of the functionality of the MacAB-like system? Is it still a resistance machinery? In my opinion this is an important point as the model is based on the truncated form of spr0693 and also used to propose a 'mode of action'.

A: To reveal the effect on the truncations of Spr0693, we performed the *in vitro* ATPase activity assays of Spr0693-0694-0695 in the liposome (Fig. 3b). Addition of the full-length Spr0693 significantly increased the ATPase activity of Spr0694-0695 to 2 folds. By contrast, addition of either the TM-truncated Spr0693 (Spr0693₃₄₋₃₉₉) or TM&MP-truncated Spr0693 (Spr0693₅₉₋₃₂₄) to Spr0694-0695 showed no significant change to the ATPase activity. The results clearly showed that the TM and MP domain of Spr0693 are essential for the full function of Spr0693-0694-0695, as well as for the resistance activity. Thus our model just shows an overall organization and architecture of the putative efflux pump, but does not represent the full-length model of Spr0693-0694-0695 in the "mode of action".

2. The identified guard helix (helix 6) should be supported by a rational mutation that destroys the interaction. What is the functional consequence of such a mutation?

A: Thank you very much for your suggestion. To test the effect of the guard helix on the function of Spr0693-0694-0695, we first tried to truncate the guard helix and express the proteins to test the effect on the ATPase activity. However, truncation of the guard helix almost completely disrupted the folding of Spr0695, as we could hardly obtain well folded protein, indicating that the guard helix is indispensable for the folding of Spr0695. Instead, we mutated three positively charged residues Lys207, Arg208, and Lys210 on the guard helix to alanine (termed *spr0693-0694-0695_{M3}*), and tested the ATPase activity. Compared to the wild-type protein, *in vitro* ATPase activity assays revealed a significantly reduced activity upon these mutations (Fig. 6d).

A previous report suggested that the antimicrobial peptide LL-37 is a putative substrate of

Spr0693-0694-0695². We performed the antimicrobial susceptibility assays, which revealed a minimum inhibitory concentration (MIC) of 8 µg/mL for the *spr0693-0694-0695* deletion strain, much lower compared to 32 µg/mL of the wild-type *S. pneumoniae* R6 strain (Table 2, Supplementary Fig. 4), further suggesting that Spr0693-0694-0695 is indeed required for *S. pneumoniae* resistance against LL-37. Complementation of the *spr0693-0694-0695* operon restores the MIC value to a level similar to the wild-type strain. However, complementation with *Spr0693-0694-0695*_{M3}, which harbors three mutations on the guard helix coding region, into *S. pneumoniae* restores the MIC to 16 µg/mL, suggesting that these mutations on the guard helix partly impair the resistance of *S. pneumoniae* against LL-37. As the flexibility of the guard helix is proposed to be important for the substrate entry, we introduced three disulfide bonds between the guard helix and the TM helices (D49C-R208C, K52C-E205C, and E374C-K207C) based on structural analysis to freeze the guard helix. Complementation with the operon harboring these mutations (termed *spr0693-0694-0695*_{3C}) results in an MIC of 8 µg/mL, similar to the *spr0693-0694-0695* deletion strain. Altogether, these results clearly suggest that the guard helix of Spr0695 is essential for the full function of Spr0693-0694-0695.

3. Line 35 / 36: ABC import systems have been described in plants as well.

A: Corrected.

4. Line 298: Protein elution was indeed performed in the absence of any imidazole?

A: Sorry. The protein was eluted with the buffer in the presence of 400 mM imidazole. We have added this statement in the Methods part.

5. Lines 310- 321: More details must be provided, otherwise no one could follow the details of the purification procedure.

A: We have added more details to describe the purification procedures in the Methods part.

6. Figure 4a: A high quality gel is required as it nearly impossible to grasp anything from the current gel.

A: We have performed the pull-down assays again and updated Figure 3a in the revised version. In addition, we performed the mass spectrometry experiments to identify each band in the SDS-PAGE, which further confirms the interaction between Spr0693 and Spr0694-0695.

Reviewer #3 (Remarks to the Author):

The authors describe a structure of the (Gram-negative) MacAB homologue from a Gram-positive bacterium, encoded by the gene cluster *spr0693-0694-0695*. The structure is derived from crystallization and using SeMet-phasing for the MacA homolog Spr0693 and using molecular replacement with the MJ0796 ATP-binding cassette protein for Spr0694-0695. The manuscript describes the structure of the Gram-positive MacAB homologue and the comparison with the *E. coli* MacAB structure and apart from a (not so convincing) pull down experiment, no additional

experiments have been made to address the fascinating questions which arise from the structure.

For the manuscript to be valuable to the reader, some of the following questions have to be addressed to make the manuscript more substantial for possible publication in Nature Communications in this referees opinion. At the moment I would not recommend publication:

1. The activity of the ATPase should be addressed. Experiments in proteoliposomes must show activity of the Spr0693-0694-0695 complex. For MacAB it has been shown that MacA stimulates ATPase activity of MacB manifold. This observation for the Spr0693-0694-0695 complex would strengthen the necessity for complex formation of a MacAB homologue from a Gram-positive bacterium.

A: Thanks for your suggestion. We have performed the *in vitro* ATPase activity assays in the liposome (Fig. 3b). The results revealed that Spr0694-0695 possesses the ATPase activity, whereas mutation of the catalytic residue or fixing the NBD dimer almost abolished the ATPase activity. Notably, addition of the full-length Spr0693 significantly augments the ATPase activity of Spr0694-0695, in agreement with that of MacAB. The results suggest that Spr0693 is essential for the full function of Spr0694-0695.

2. The guard helix appears to be an interesting and important feature as observed from the structure. However, it is not clear what the helix is guarding. From Fig. 5a it is not evident that it blocks the lateral entrance. Since the guard helix is rich in positive side chains, as the putative substrate LL-37, it would be very helpful to have some structure/function activity relationship experiments, including variants with substituted side chains in the guard helix.

A: Thanks for your suggestion. We have performed the ATPase activity assays (Fig. 6d) and the antimicrobial susceptibility assays in *S. pneumoniae* R6 against the putative substrate LL-37 (Table 2). All the results revealed that the guard helix is essential for the full function of Spr0693-0694-0695. For more details, please see the response to the Question 2 raised by Reviewer 2.

3. The manuscript is also in need for a morphing (movie) between the postulated resting state of the Spr0694-0695 complex and the postulated occluded state of MacB.

A: Following your suggestion, we have added a couple of sentences to describe the movie between the two states. We speculate that our Spr0694-0695 structure more likely represents a rest state. In this state, the two NBDs adopt a somewhat closed conformation but yet accessible for ATP; however, the substrate entrance is blocked by the guard helix $\alpha 6$. At this moment, Spr0693 does not associate with Spr0694-0695, considering the relatively unstable complex of Spr0693-0694-0695 in solution. Upon approaching of the substrate, the hexameric Spr0693 packs against the Spr0694-0695 complex. The PLDs of TMD rotate outwards and the guard helix shifts away to form a continuous substrate-entrance tunnel. Thus the structure of MacAB-TolC in complex with an unknown molecule should represent a substrate/inhibitor occluded state. Binding and hydrolysis of ATP in NBDs will trigger a series of conformational changes that are required for the substrate transport, which is supported by the ATPase activity assays of Spr0694 G173C mutant.

4. The complex formation data are not very convincing. After Ni-NTA purification, assuming only

Spr0694-0695 His-tagged protein in the solubilized material, how does the SDS-PAGE analysis look like? The gel shown only shows the result after Ni-NTA and StrepII-tag pull-down. Is the complex stable in e.g. SEC? The stained gel in Fig. 4a shows also shows a small amount of protein loaded. The pull-down needs to contain proper controls, like a pull-down with non-tagged complex and a positive control with e.g. MacAB from *E. coli*. Data could be substantially supported by MS.

A: We have performed the pull-down assays again with negative controls (the revised Fig. 3a). The eluted proteins were separated by electrophoresis and the corresponding bands were further detected by mass spectrometry (Supplementary Fig. 1). The results showed that the elution contains all the three components, indicating a complex of Spr0693-0694-0695 in vitro (Fig. 3a). Notably, only a small fraction of the input prey protein could be pulled down, suggesting the complex is relatively unstable, which is in agreement with the heterogeneous profiles of size-exclusion chromatography (Supplementary Fig. 2).

5. In Fig 4a, one of the bands is designated to a truncated version of Spr0695. How do the authors reach that conclusion? Mass-spec data would help to verify the identity of the band in the gel.

A: Indeed, we have performed mass spectrometry and confirmed this band is a truncated version of Spr0695 (Supplementary Fig. 1).

6. Is the MacA homologue Spr0693 capable to interact with TolC in *E. coli*. Does expression of the Spr0693-0694-0695 complex in *E. coli* decrease susceptibility against e.g. Erythromycin, fusidic acid, chloramphenicol?

A: Thanks for your suggestion. We first analyzed the interaction pattern of TolC and its partner proteins, and found that the interaction between TolC and its partner proteins MacA or AcrA are mostly via van der Waals interactions, mainly contributed by bulky residues, such as glutamine and lysine. However, the loop regions at the outermost moiety of Spr0693 are mainly composed of residues of small side chain (APVGGEDATVQ), which do not seem to interact with *E. coli* TolC. In addition, the TolC homolog is absent from *S. pneumoniae* or other Gram-positive bacteria that lack the outer membrane. Furthermore, we compared the susceptibility of *E. coli* against either chloramphenicol or erythromycin with or without overexpression of Spr0693-0694-0695. The results showed that overexpression of Spr0693-0694-0695 did not increase the resistance capability of *E. coli* against antibiotics. Altogether, we propose that Spr0693-0694-0695 might not need a TolC homolog to fulfill its function.

7. Is it possible to express the Spr0693-0694-0695 complex in its cognate host? If yes, a pull down of this complex might reveal other proteins attached. This could reveal more on the functional role, since it is puzzling why spr0694-0695 is expressed in an operon with spr0693. The reason to have such a complex in Gram-positive bacteria is one of the main questions.

A: Based on the bioinformatic analysis using Prokaryotic Operon DataBase³, *spr0693* and *spr0694-0695* are predicted to be in a single operon. In addition, the reverse transcription PCR experiments also confirmed that the *spr0693* and *spr0694-0695* genes are co-transcribed into a single mRNA molecule. In addition, the ATPase activity assays and antimicrobial susceptibility assays also

suggest that the intact Spr0693-0694-0695 complex is required for the full function of the putative efflux pump. Thus it is reasonable to speculate that the Spr0693-0694-0695 complex forms a channel to transport the substrate from the plasma membrane into the peptidoglycan layer of *S. pneumoniae*.

8. The change in the conformation of the NBD domains between the resting and occluded states could be analyzed via Cys-crosslinking (single Cys variants have to be constructed). It verifies the structure and gives insight in the conformational changes during the ATP hydrolysis cycle, which is still elusive for this TM8-fold group of transporters compared to the other ABC transporters.

A: Thanks for your suggestion. Structural analysis revealed that Gly173 of Spr0694 is located at the dimeric interface of NBDs and has a distance of 8 Å with its counterpart of the symmetric subunit. To test whether the conformational changes of NBDs are also occurred during ATP hydrolysis in Spr0694-0695, we thus mutated Gly173 of Spr0694 into cysteine, and made a cross-linking experiment at the oxidative condition. Electrophoresis profiles clearly confirmed that the G173C mutant of Spr0694 forms an intermolecular disulfide bond in solution in the presence of Cu²⁺ (Supplementary Fig. 3), which means that the NBD dimer is covalently fixed in a closed state. Consequent ATPase activity assays showed that this mutant almost completely abolished the activity (Fig. 3b), which confirms the ATP hydrolysis cycle is also essential for this TM8-fold group of ABC transporters, similar to that of other ABCs.

9. The flexible N211-V224 loop in Spr0693 is said to be non-conserved and this would be an indication for a rather broad spectrum of substrate selectivity. I cannot follow this line of argumentation. The other hypothesis for this loop to be a valve preventing backflow, is already difficult to follow in a MacAB-TolC setup, but in a MacAB like setup as suggested for the Spr0693-0694-0695 complex seems not to make much sense, since the substrates are exported in the medium anyway, whether they go through spr0693 or are only transported by Spr0694-0695 and end up in the outside medium as well.

A: We have systematically analyzed the operon organization of *spr0693-0694-0695* homologs in other species and found that all members share a conserved gene organization with the co-transcription of these three genes in a single operon. Our ATPase activity assays and antimicrobial susceptibility assays also suggest that the intact Spr0693-0694-0695 complex is required for the full function. Considering the sequence variations of the gating ring loop in Spr0693, we propose that this loop might not determine the substrate specificity, but instead function as a valve to prevent backflow, similar to that in MacAB. However, we cannot rule out the possibility that some given substrates can be specifically exported into the medium directly without going through Spr0693. More investigations are needed to identify the real substrates and to elucidate the fine transport mechanism of this group of ABCs.

10. I wonder if the docking of Spr0693 to Spr0694-0695 makes much sense, since the ECDs have a different conformation. Is it anyway likely that Spr0693 and Spr0694-0695 are forming a complex at all times?

A: Based on the limited data, the model presented here has some defects as mentioned in the manuscript. However, the model indeed gives us hints for the assembly of this group of ABC transporters in Gram-positive bacteria. Of note, the conformational changes of ECDs are coordinated

with the transport of substrates, but might not affect the overall organization of the complex. The size-exclusion chromatography and pull-down assays also suggest that Spr0693-0694-0695 forms a complex in vitro. However, the complex is not very stable, indicating that they are not forming a complex at all times, as you proposed.

11. Is Spr0693 attached to the membrane? Is complex formation and disassembly of the complex thought to be transient?

A: Indeed, sequence analysis revealed that Spr0693 contains a single transmembrane helix, which helps Spr0693 anchor to the membrane. Purification of the full-length Spr0693 also needs extraction of the proteins from the membrane, further confirming that Spr0693 is a membrane protein. The size-exclusion chromatography and pull-down assays suggest that Spr0693-0694-0695 complex is not very stable in vitro, indicating that this complex should be transient.

12. Do the authors have a suggestion why a Gram-positive organism needs to have a bipartite setup including a nanotube channel protein?

A: Gram-positive bacteria have only one layer of membrane but have a thick layer of peptidoglycan. Recently, using Cryo-EM, Zuber, B. et al have found a layer of low electron density between the plasm membrane and peptidoglycan layer in Gram-positive bacteria, which is termed inner wall space⁴. The distance between the apex of the Spr0693-0694-0695 assembly and the extracellular face of the membrane is about 154 Å, which is comparable to the reported thickness of inner wall space. The nanotube channel may serve as a tunnel to transport the substrate into the peptidoglycan layer, probably required to avoid direct access of these substrates/toxins to the inner membrane.

The last question, the role of the guard helix, and the stability of the bipartite complex should be supported with functional data to underpin the structural information the authors obtained from the structure. The structure alone is interesting, but we do not learn very much on the structure/function relationship of this group of transporters within the ABC transporter family.

A: To have support from functional data, we performed the antimicrobial susceptibility assays against LL-37 in *S. pneumoniae* R6, with the emphasis on the role of the guard helix. In addition, we further detected the complex formation and stability using pull-down assays, size exclusion chromatography, together with mass spectrometry.

References

1. Tikhonova, E.B., Devroy, V.K., Lau, S.Y. & Zgurskaya, H.I. Reconstitution of the *Escherichia coli* macrolide transporter: the periplasmic membrane fusion protein MacA stimulates the ATPase activity of MacB. *Mol. Microbiol.* **63**, 895-910 (2007).
2. Majchrzykiewicz, J.A., Kuipers, O.P. & Bijlsma, J.J.E. Generic and Specific Adaptive Responses of *Streptococcus pneumoniae* to Challenge with Three Distinct Antimicrobial Peptides, Bacitracin, LL-37, and Nisin. *Antimicrob. Agents Chemother.* **54**, 440-451 (2010).
3. Taboada, B., Ciria, R., Martinez-Guerrero, C.E. & Merino, E. ProOpDB: Prokaryotic Operon DataBase. *Nucleic Acids Res.* **40**, D627-D631 (2012).
4. Zuber, B. *et al.* Granular layer in the periplasmic space of Gram-positive bacteria and fine structures of *Enterococcus gallinarum* and *Streptococcus gordonii* septa revealed by cryo-electron microscopy of vitreous sections. *J. Bacteriol.* **188**, 6652-60 (2006).

Reviewers' Comments:

Reviewer #1:

Remarks to the Author:

The authors addressed the criticism constructively and the revised manuscript is significantly improved. A few issues need to be addressed. Specifically:

1. Figs 3 (all panels) and 6D. The labels on the graphs are impossible to read. The meaning of the error bars and number of experiments are not described in the legends.
2. There is no statistical analyses. Authors should avoid using the word "significant difference" in the description of ATPase activities if no statistical analyses were carried out to demonstrate the significance of less than two fold difference.
3. Since authors show ATPase activities, some basic kinetic parameters should be reported.

Reviewer #2:

Remarks to the Author:

All points raised have been properly addressed in the revised version of the manuscript. Thus, I recommend accept as it is.

Reviewer #3:

Remarks to the Author:

I would like to thank the authors for the additional experiments they have conducted. These are really helpful and underpins the hypotheses made on the assembly and the role of the guard helix.

Two questions still left:

1. What was the specific ATPase activity measured? How does this relate to the activities measured by other ABC transporters (incl. MacAB-ToIC)? Could the authors indicate the specific activity (Pi produced/mg/min). How long was the activity measured for (to include into lines 475-478).
2. Out of interest, I would like to ask why the ATPase activity test was initiated by adding ATP, followed by freeze/thaw cycles? I would assume that the reconstitution would yield Spr0694 and Spr0693 subunits at both sides of the membrane. However, Pi formation and detection can only be detected from the outside of the proteoliposomes? Or was the sample solubilized once more after the reaction and both Pi generated at inside and outside of the proteoliposomes was measured?

REVIEWERS' COMMENTS:

Reviewer #1 (Remarks to the Author):

The authors addressed the criticism constructively and the revised manuscript is significantly improved. A few issues need to be addressed. Specifically:

1. Figs 3 (all panels) and 6d. The labels on the graphs are impossible to read. The meaning of the error bars and number of experiments are not described in the legends.

A: We have adjusted the labels in Figure 3 and Figure 6d. The data are presented as the means \pm S.D. from three independent assays. We have added this information in the legends.

2. There is no statistical analyses. Authors should avoid using the word "significant difference" in the description of ATPase activities if no statistical analyses were carried out to demonstrate the significance of less than two fold difference.

A: We have indeed performed statistical analyses using the two-tailed Student's *t*-test. Accordingly, we have added the information in the revised legends.

3. Since authors show ATPase activities, some basic kinetic parameters should be reported.

A: Thanks a lot. Following your suggestion, we have adopted the V_{\max} of the ATPase activity, instead of relative activity, in our revised figures (Figure 3b and Figure 6d) and legends.

Reviewer #2 (Remarks to the Author):

All points raised have been properly addressed in the revised version of the manuscript. Thus, I recommend accept as it is.

Reviewer #3 (Remarks to the Author):

I would like to thank the authors for the additional experiments they have conducted. These are really helpful and underpins the hypotheses made on the assembly and the role of the guard helix.

Two questions still left:

1. What was the specific ATPase activity measured? How does this relate to the activities measured by other ABC transporters (incl. MacAB-TolC)? Could the authors indicate the specific activity (Pi produced/mg/min). How long was the activity measured for (to include into lines 475-478).

A: The ATPase activity was measured by the Pi produced as the consumption of ATP. We have indicated the activities by calculating the Pi produced in mole with the consumption of ATP by per mole of Spr0694-0695 protein per minute, which was also adopted for other ABC transporters, including MacAB-TolC (Tikhonova, Devroy et al. 2007). The activity was measured after reaction for 30 min, which have been added in the revised Methods.

2. Out of interest, I would like to ask why the ATPase activity test was initiated by adding ATP, followed by freeze/thaw cycles. I would assume that the reconstitution would yield Spr0694 and Spr0693 subunits at both sides of the membrane. However, Pi formation and detection can only be detected from the outside of the proteoliposomes? Or was the sample solubilized once more after the reaction and both Pi generated at inside and outside of the proteoliposomes was measured?

A: The liposomes could easily aggregate or overlap with each other during the experiment, thus a freeze/thaw cycle before initiating the reaction would help to expose proteins reconstituted into proteoliposomes to fully contact with the substrates. This procedure was also used in the ATPase activity assay of MacAB (Tikhonova, Devroy et al. 2007) and other transporters (Taylor, Manolaridis et al. 2017). We agree with you on the opinion that the reconstitution would yield Spr0694 and Spr0693

subunits at both sides of the membrane; however, only Pi produced outside of the proteoliposomes could be detected.